# Insight into Electrical and Dielectric Relaxation of Doped Tellurite Lithium-Silicate Glasses with Regard to Ionic Charge Carrier Number Density Estimation

**DOI:** 10.3390/ma13225232

**Published:** 2020-11-19

**Authors:** Young Hoon Rim, Chang Gyu Baek, Yong Suk Yang

**Affiliations:** 1College of Liberal Arts, Semyung University, Chechon, Chungbuk 27136, Korea; 2College of Nanoscience and Nanotechnology, Pusan National University, Busan 46241, Korea; newbrave@pusan.ac.kr

**Keywords:** Li_2_O-2SiO_2_-TeO_2_, power law, Cole–Cole plot, ionic conductivity, activation energy

## Abstract

We investigate the role of tellurite on a lithium-silicate glass 0.1 TeO_2_ − 0.9 (Li_2_O-2SiO_2_) (LSTO) system proposed for the use in solid electrolyte for lithium ion batteries. The measurements of electrical impedance are performed in the frequency 100 Hz–30 MHz and temperature from 50 to 150 °C. The electrical conductivity of LSTO glass increases compared with that of Li_2_O-2SiO_2_ (LSO) glass due to an increase in the number of Li^+^ ions. The ionic hopping and relaxation processes in disordered solids are generally explained using Cole–Cole, power law and modulus representations. The power law conductivity analysis, which is driven by the modified Rayleigh equation, presents the estimation of the number of ionic charge carriers explicitly. The estimation counts for direct contribution of about a 14% increase in direct current conductivity in the case of TeO_2_ doping. The relaxation process by modulus analysis confirms that the cations are trapped strongly in the potential wells. Both the direct current and alternating current activation energies (0.62–0.67 eV) for conduction in the LSO glass are the same as those in the LSTO glass.

## 1. Introduction

Understanding lithium ion conduction provides a broad application opportunity ranging from the small size of portable devices to large size vehicles and electric energy storage systems as lithium ion secondary batteries consist of a cathode, an anode and electrolyte. Concerning the electrolyte of a lithium ion battery, in most cases, the host material of lithium ion conduction can be a liquid, polymer or solid. Up to now, in the sense of applications, a liquid and polymer have been used widely due to the easy fabrication process and high ionic conductivity. Nonetheless, with the enormous increase on the demand for lithium ion batteries, higher safety and longer lifetime have been required. All solid-state batteries, where the components consisting of the batteries are solid, have turned out to be the best candidates to fulfill those conditions. Thus, studies have been concentrated on developing solid state materials for the use of a cathode, an anode and electrolyte of a lithium ion battery [1,2,3,4,5,6].

Regarding solid-state electrolytes, material types can be crystal or glass. Among these, the glass material has its own advantages characterized by the possible selection of various atomic species, easy control of synthesizing temperature by choosing proper glass former, easy variation of doping rates and types for improving electric properties and lack of directional malfunctioning [7]. Lithium-silicates, as forms of ceramics, have long been studied because of useful applications such as dental materials and coating binders. Meanwhile, lithium-silicate glasses with a high concentration of lithium have been recognized to be used for solid state electrolytes of a lithium ion battery. One of the barriers to overcome in using solid-state materials for an electrolyte is a relatively low electrical conductivity. 

There have been continuous efforts to develop crystalline solid electrolytes based on traditional materials. Those are a lithium superionic conductor (LISICON), which normally refers to the chemical formula Li_2+2x_Zn_1−x_GeO_4_, and LISICON-like conductors such as Li_10_MP_2_S_12_ (M = Si, Ge, Sn), exhibiting ionic conductivity of about 10^−6^ and 10^−2^ Scm^−1^ at room temperature, respectively [8]. In the sense of extending conducting ions, sodium superionic conductor (NASICON) with the normal chemical formula Na_1+x_Zr_2_Si_x_P_3−x_O_12_ and the NASICON-like conductor such as Li_1.3_Al_0.3_Ti_1.7_(PO_4_)_3_, are also widely studied materials, but their ionic conductivities are one order lower compared with those of LISICON and LISICON-like materials due to the heavier mass of conducting ions [8]. 

Solid polymers of polyesters, polyamines and polysiloxane are also candidates for electrolytes. Ions are usually conducted through the polymer chains. Polymers are easy to process and advantageous for large scale production, high plasticity and elasticity. Considering the low limit of conductivity 10^−4^ Scm^−1^ for solid electrolytes, most polymer electrolytes are within this range of conductivity, but the high limit of polymer is confined to about 10^−3^ Scm^−1^ at room temperature [9]. 

Concerning amorphous solid electrolytes under development, there can be two types of non-oxide and oxide ceramic glasses. Li_2_S-GeS_2_, Li_2_S-P_2_S_5_ with conductivity about 10^−4^ Scm^−1^ and Li_2_S-SiS_2_-Li_3_PO_4_, Li_2_S-SiS_2_-Li_4_SiO_4_ with conductivity about 10^−3^ Scm^−1^ are typical ceramic glass electrolytes which have high ionic conductivities comparable to those of the LISICON and NASICON series [10,11].

Tellurium is one of the chalcogens and, when it is doped or added to a host, the mechanical, electrical, optical properties of material can be controlled. It vulcanizes rubber, changes electronic current in semiconductors and tints crystal or glass color. In the form of its oxide, the tellurite acts as a glass former and leads tellurite mixed materials to be disordered in their network structure [12]. Therefore, as is the case in this work, the addition of tellurium or tellurite in an oxide material allows us to adjust both electrical conductivity and glass forming ability.

Recently, Rodrigues et al. investigated the electrical conductivity of 0.30 Li_2_O − 0.70 (x TeO_2_ − (1 − x) SiO_2_) glass [13] and found that the electrical conductivity increased as they increased the tellurium oxide content to x = 0.25. They reported that their system showed a lithium ionic conductivity of 1.78 × 10^−6^ S/cm at 150 °C.

We observe the consistent results that the electrical conductivity of the 0.9 (Li_2_O − 2SiO_2_) − 0.1 TeO_2_ (LSTO) glass increases in comparison with the conductivity of the Li_2_O-2SiO_2_ (LSO) glass, and the further investigation is needed to answer for the fundamental questions on the origin of ionic conduction mechanism. 

Recently, we showed that with the modified fractional Rayleigh equation, the ionic carrier number density in the glass system can be estimated [14]. 

In this paper, we aim to find out the effects of doped tellurium oxide for Li ionic conductivity in the silicate glasses by comparing the electrical impedance of LSO and LSTO glasses. We analyze the electrical characteristics using the Cole−Cole plot, power law and modulus representations to unveil the differences in the ionic conductivity between LSO and LSTO glasses. 

## 2. Experimental

The binary and ternary compositions, x (Li_2_O − 2SiO_2_) − (1 − x) TeO_2_ (LSO; x = 1, LSTO; x = 0.9 mol ratio), were first prepared using Li_2_CO_3_ (Sigma-Aldrich, 99%, USA), SiO_2_ (Sigma-Aldrich, 99.5%, USA) and TeO_2_ (Sigma-Aldrich, 99%, USA). Those three chemicals were thoroughly mixed with the mole ratios of 1:2:0 for a binary sample and 0.9:1.8:0.1 for a ternary sample for one hour. 

Each mixture was melted at a high temperature of 1300 °C to eliminate CO_2_ by evaporation. We used a high temperature endurable platinum container on melting chemicals during the sample preparation procedure to avoid any reaction between compounds and the crucible. For the synthesis of an amorphous solid, a cooling rate from the melt is one of the critical factors, and a proper quenching condition has to be chosen within a certain range. Many time trials with different cooling rates are normally required to obtain a homogenous glass. 

We used the self-made thick copper plates to fabricate glass samples. Pouring the melt between the copper plates allowed the sample to be less than 1 mm thick film with a cooling rate of about 10^3^/s. Glass characteristics of the fabricated samples were checked with the broad patterns scattered from the disordered network structure in X-ray diffraction (XRD, λ = 1.5406 Å, Miniflex II, Rigaku, Tokyo, Japan) patterns, and with the exothermic peaks of enthalpy change accompanying the first order phase transition from a glass to a crystal in differential thermal analysis (DTA, 2000 s, Japan) patterns. 

For the impedance studies, because the real measurements are related to the pairs of resistance-capacitance and the capacitance of the sample is very sensitive to the thickness and surface uniformity, an additional sample preparation is required. For this purpose, each bulk glass sample was melted again at 1300 °C in a platinum container and dropped between two rotating copper rollers, which enabled the fast cooling rate, uniform surface and thinner film of 0.5 mm. 

For XRD measurements, we used a step scan method with the scattering angle increase of 0.05° for three seconds. DTA measurements were carried out under both nitrogen and air environments with the gas flow rate 100 cc/min. 

In order to obtain the electrical properties of the conductivity, impedance and modulus of LSO and LSTO glasses, the impedance gain/phase analyzer (4294A, Keysight, USA) was used. On both sides of the surface of each glass sample synthesized with the twin roller quenching mentioned above, the gold circle film of 1.5 mm was coated as an electrode by the evaporation method. The electrical measurement conditions were the heating of 5 °C/min, the frequency range 100 Hz–30 MHz, under an air environment.

## 3. Results and Discussion

Figure 1a,b show the x-ray diffraction patterns of the Li_2_O-2SiO_2_ (LSO) and 0.9 (Li_2_O-2SiO_2_) − 0.1TeO_2_ (LSTO) samples at room temperature. The Figure exhibits the broad peaks of the amorphous phase. The broad XRD pattern is the typical glass characteristics scattered from the short ranging disordered network structure. The insets of Figure 1a,b show the non-isothermal differential thermal analysis (DTA) curve for the LSO and LSTO glasses with the heating rate 10 °C/min. From the inset of DTA curves, the glass transition temperatures are determined to be 461 °C for LSO glass and 437 °C for LSTO glass. It is observed in the LSTO glass that T_g_ and crystallization temperature T_c_ decrease if TeO_2_ is introduced into the LSO glass, indicating that the bonding strength of the structural network is weakened. Comparing this with the density of LSO, meanwhile, it is known that the density of LSTO increases [13].

For non-Debye complex systems, the Cole−Cole representation is useful to figure out the dynamic relaxations in the amorphous solids [15].
(1)$$Z^{*}=\frac{\Delta R}{1+{\left(i\omega \tau \right)}^{\delta }}\times \frac{A}{L}$$
where $\Delta R≅R_{0}-R_{\infty }$, $R_{0}$ is the resistance at zero frequency, $R_{\infty }$ is the resistance at infinity of frequency, L is the sample thickness and A is the electrode area of sample. The exponent parameter  δ is related to the distribution of potential barriers, which characterize the dielectric relaxation of the ions in the sample. The parameter δ varies in the range of 0≤δ≤1.

The temperature dependence of the Cole−Cole plots for (a) LSO glass and (b) LSTO glass is shown in Figure 2. A depressed circular arc intersects at the Z’ axis of the low-frequency region, which gives the direct current (dc) conductivity, $\sigma_{\mathrm{dc}}=L/\left(Z_{0}A\right)$, at given temperatures. For example, the measured value of *Z_0_* = 30.2 MΩcm for LSO glass and *Z_0_* = 25.1 MΩcm for LSO glass at T = 50 °C, indicating that $\sigma_{\mathrm{dc}}\left(\mathrm{LSTO}\right)/\sigma_{\mathrm{dc}}\left(\mathrm{LSO}\right)$ = 1.20. The results of electrical measurements show that the electrical conductivity increases for adding the element TeO_2_ into the lithium-silicate glass. The result shows a good agreement with the result obtained by Rodrigues et al. [13].

The exponent δ approaches 1 for the ideal Debye system and the exponent decreases if the structural disorder increases. The exponent δ in Equation (1) for a bulk sample varies from 0.78 to 0.85 for LSO glass and it varies from 0.77 to 0.82 for LSTO glass in the temperature range from 50 °C to 150 °C. The Cole−Cole representation shows that the TeO_2_ content in LSTO glass slightly reduces the variation of the distribution of potential barriers, resulting in the fact that the distribution of energy barriers is relatively narrow for LSTO glass. We remind readers of the fact that the structural network is modified, and the strength of bonds are to be weakened if the TeO_2_ is added in LSO glass because of the mixed glass former effect (MGFE). The MGFE is a known method to increase the ionic conductivity by changing the ionic pathways, in which the saddle point energies for ion jumps between neighboring non-bridging oxygen (NBO) sites [16,17,18,19,20]. 

Comparing the variation of the exponent δ for both samples, we expect that the glass former effect is small for LSTO glass. As an increasing temperature, the exponent δ increases in the LSO and LSTO glasses. Thus, the structural disorder decreases for both glasses as an increasing temperature. 

Figure 3 shows the variation of dc and alternating current (ac) conductivities of LSO and LSTO glasses in terms of reciprocal temperature dependence. The dc and ac conductivities in Figure 3 obey the Arrhenius relations $\sigma_{dc}T=\mathrm{Cexp}\;\left(-\frac{E_{\mathrm{dc}}}{k_{B}T}\right)$ and $\omega_{p}=\omega_{0}\exp\;\left(-\frac{E_{\mathrm{ac}}}{k_{B}T}\right)$. Here, E_dc_ is the dc activation energy and E_ac_ is the ac activation energy, $\omega_{0}$ is the onset frequency, $\omega_{p}$ is the peak frequency, C is a constant and $k_{B}$ is the Boltzmann constant. The conductivities and activation energies are calculated from the complex impedance Cole−Cole representation in Equation (1). The peak frequency, $\omega_{p}$, is a maximum value of the imaginary part of the Cole−Cole plot and it satisfies the relation to the relaxation time such as $\omega_{p}\tau =1$ [15]. The values of the activation energies can be obtained by the slopes from the least-squares straight-line fits to the data. 

The Arrhenius equation implies the fact that the charge carriers are thermally activated, and the dc and ac conductivities vary linearly with temperature. For LSO glass, E_dc_ = 0.67± 0.02 eV and E_ac_ = 0.66 ± 0.02 eV. Whereas, for LSTO glass, E_dc_ = 0.66 ± 0.04 eV and E_ac_ = 0.66 ± 0.01 eV. The values of the dc and ac activation energies of LSO glass is the same as the values of LSTO glass. The results indicate that the average potential barriers for hopping in the LSO and LSTO glasses are the same in the local area of network as well as in the whole network medium. The distribution of energy barriers in the LSO glass is slightly broadened to compare with the one in the LSTO glass, indicating that the relaxation times in the LSO glass are relatively short compared with those in the LSTO glass. In other words, the ac hopping frequency, $\omega_{p}$, in LSO glass is slightly faster than the one in LSTO glass, as seen in Figure 3. 

Kunow and Heuer pointed out the fact that the influence of non-bridging oxygens (NBOs) for local network fluctuations in the silicate glass was most important factor for the lithium dynamics. They showed in computer simulation that the conductivity increased with an increase of NBOs sites [21]. The tetrahedral units of SiO_4_ were broken into different numbers of NBO atoms for every silicon atom by the network modifier, Li-ion, and the sites of the NBOs in the lithium-silicate glasses were created [19]. When the new NBOs sites were formed via breaking of Si-O bonds by the Li-ions, the number of hopping sites would increase. Meanwhile, it was reported that the ionic conductivity of lithium oxide glass exhibited a low conductivity due to a strong trap of NBO atoms for lithium ions [22]. The addition of TeO_2_ in the Li_2_O-SiO_2_ glass results in a weakened Coulomb interaction within the structural network of the modified glasses, revealing that the electrical conductivity increases due to an increase in the number of charge carriers. We will mention it later.

Rim et al. derived the following expression of the ac conductivity spectra of glasses that it consists of a superposition of a Jonscher term and a linear dependent frequency term [14], i.e.,
(2)Reσ(ω)=σdc+Kωs+Aω = σdc[1+(ωωh)s]+Aω

Here, $\sigma_{\mathrm{dc}}$ is the conductivity at the frequency ω=0, $K\sim \frac{\mathrm{Nq}}{\sqrt{2\pi }}\sum_{n=1}^{\infty }\frac{{b\eta }_{\alpha }a^{n-1}\Gamma \left[1+\left(n-1\right)\alpha \right]}{\Gamma \left(n\right)}\sin\left(\frac{n\alpha }{2}\pi \right)$, the exponent *s* varies in the range from 0 to 1, 0 < s=1−nα < 1 and the constant A is related to the upper limit of characteristic time $\tau_{c}$, n is an integer number. The hopping frequency $\omega_{h}$ is defined by Reσ(ωh)=2σdc empirically. 

In the dispersive region, the second term, ${K\omega }^{s}$, in Equation (2) contributes to the total ac conductivity caused by displacement of the cations, where the dispersive region sets in approximately at the onset frequency $\omega_{h}$. The last term, Aω, in Equation (2) is the constant loss term. The results from the modified fractional Rayleigh equation indicate that the mobile ions are hopping through the fractal structural percolation pathway [14].

At various temperatures, Figure 4a,b show the frequency spectra of the real conductivity Reσ(ω) for LSO and LSTO glasses. The dc conductivity of LSTO glass slightly increases compared to the one of LSO glass at various temperatures.

The value of power law exponent *s* of LSO glass varies from 0.61 to 0.72 at different temperatures but the exponents of LSTO glass are dispersed from 0.57 at 50 °C to 0.69 at 150 °C. The exponents are the fit from Equation (2) at several temperatures. Based on the modified fractional Rayleigh equation, we have obtained one of the results that the conduction ions are moving through the first branch of the fractal structured pathways [14].
(3)$$s=\frac{4}{3}-\frac{1}{D_{f}}$$

Consequently, the Li-ions of LSO and LSTO glasses are moving through the fractal pathway with the dimensions D_f_ ≈ 1.3~1.6 in lithium-silicate glass systems, which depends on the temperatures. 

Combining the measured values of the exponent *s* and the hopping frequency ω_h_, we can determine the number of conducting cations per mole, which is expressed by $\rho_{m}=\frac{1}{\sqrt{M}}\omega^{3\left(1-s\right)}$ [14]. For example, the obtained number of carriers per mole are $\rho_{m}\left(\mathrm{LSO}\right)$ = 2.72 × 10^16^ and $\rho_{m}\left(\mathrm{LSTO}\right)$ = 2.59 × 10^17^ at T = 50 °C. Meanwhile, we obtain the number densities $\rho_{m}\left(\mathrm{LSO}\right)$ = 7.80 × 10^16^ and $\rho_{m}\left(\mathrm{LSTO}\right)$ = 7.70 × 10^17^ at T = 150 °C. For calculation, we use the atomic mass M(LSO) = 150.0 g/mol and M(LSTO) = 151.0 g/mol and the Avogadro’s number N_A_ = 6.02 × 10^23^/mol. The measured values of LSO glass are exponent s = 0.61 and the hopping frequency $\omega_{h}$ = 6.52 × 10^4^ rad/s at T = 50 °C, while the measured values of LSTO glass are exponent s = 0.57 and the hopping frequency $\omega_{h}$ = 1.34 × 10^5^ rad/s at T = 50 °C. 

It is interesting to point out that the estimated cation concentration ratios of $\rho_{m}\left(\mathrm{LSTO}\right)/\rho_{m}\left(\mathrm{LSO}\right)$ are equal to 9.52 at T = 50 °C and 9.87 at T = 150 °C but the observed dc conductivity ratios of $\sigma_{\mathrm{dc}}\left(\mathrm{LSTO}\right)/\sigma_{\mathrm{dc}}\left(\mathrm{LSO}\right)$ are equal to 1.37 at T = 50 °C and 1.38 at T = 150 °C, showing that the ratios are almost independent of temperature. Thus, the decreasing ratio of $\sigma_{\mathrm{dc}}/\rho_{m}$ is about 0.14 at a couple of temperatures, indicating that the partial number of ions, about 14%, in LSTO glass contribute to increase the dc conductivity of LSTO glass compared with the one of LSO glass. The ionic conductivity is hindered as a consequence of the trapping in the non-bridging oxygen sites and the scattering process in the glass system. Again, the results indicate that the electrical conductivity of LSTO glass increases because of an increase in the number of Li-ions. However, it also reveals that only partial cations may contribute to ionic conductivity. 

We note that the observed dc conductivity ratio of LSTO/LSO from the power law formula is almost the same as the one from the Cole−Cole plot. The incorporation of tellurite units into a silicate network forms a net with the silicate chains and creates a more cross-linked network, leading to the fact that a structural network is more suitable for ion trapping. The mechanisms that govern the ion conduction in the inorganic solid-state electrolytes for lithium batteries show the inter-relationship between ion size and lattice volume. That is, the lithium ion conductivity within the LISICON-like electrolytes can be increased by several orders of magnitude when the lattice volume increases by substitution elements, which acts to reduce the activation energies of the LISICON-like electrolytes. On the contrary, the conductivity can be decreased in the following cases. Firstly, if the ions are too large in comparison with the lattice constant then the diffusion of ions are limited by moving through structural bottlenecks in the pathways. Secondly, if the ions are too small then they become trapped in a potential minimum [8]. Therefore, a decrease of the dc conductivity of LSO and LSTO glasses resulted in the trapping of lithium ions within the hopping NBOs sites of Si-O bonds, suggesting that the lithium ions are too small in comparison with the spacious hopping sites in the network [19]. 

A temperature-dependent Arrhenius plot is represented in Figure 5. We have obtained it from the fit of the Jonscher’s power law in Equation (2). The straight lines in the figure give the values of the conductivity activation energies. For LSO glass, E_dc_ = 0.67 ± 0.01 eV and E_ac_ = 0.66 ± 0.03 eV. For LSTO glass, E_dc_ = 0.67 ± 0.01 eV and E_ac_ = 0.67 ± 0.01 eV.

The values of activation energies are almost identical for both the Jonscher’s conductivity and Cole−Cole impedance, indicating that the conduction mechanism is in a good agreement with the electrical relaxation.

The modulus representation can provide information about the relaxation mechanism [23,24]. The relationship between the modulus $M^{*}\left(\omega \right)$ and dielectric constant $\varepsilon^{*}\left(\omega \right)$ is
(4)$$M^{*}\left(\omega \right)\;=\;1/\varepsilon^{*}\left(\omega \right)\;=\;M^{\prime }\left(\omega \right)+{\mathrm{iM}}^{\prime \prime }\left(\omega \right)$$

Figure 6 shows the normalized imaginary part of electrical modulus, $M^{\prime \prime }\left(\omega \right)/M^{\prime \prime }{\left(\omega \right)}_{\max}$ versus log ($\omega /\omega_{\max}$) for LSO and LSTO glasses at various temperatures. The low frequency region below $M^{\prime \prime }{\left(\omega \right)}_{\max}$ is associated with the hopping conduction over long distances. In high frequencies over $M^{\prime \prime }{\left(\omega \right)}_{\max}$, ions move back and forth inside of potential barriers in short distances [23,24,25,26].

The deviation from the master curve for LSO and LSTO glasses at 50 °C is observed in low and high frequency regions. However, the deviation is immerged into the master curve at 130 °C in a high frequency region, indicating that the cations in LSTO glass are confined to the potential wells as much as those in LSO glass at a high temperature. It is also observed for LSO and LSTO glasses that the full width at half-maximum (FWHM) slightly decreases with the increasing temperature, suggesting that the interaction between conducting ions and surrounding NBOs slightly increases for an increasing temperature. 

On the other hand, the FWHM’s of LSTO glass broadens by about 0.65 decades compared with the FWHM’s of LSO glass especially at high frequencies, exhibiting that the cations in LSTO glass are trapped strongly in the potential wells [26]. As already mentioned, the big change of the FWHM of LSTO glass may be due to the loosened network for the glass containing TeO_2_. Therefore, we may conclude that the relaxation process and ac conductivity are in close agreement. 

Figure 7 shows the plot of relaxation frequency, $\ln\omega_{m}$ versus 1/T is represented by the modulus formula. From the slope of the $\ln\omega_{m}$ vs. 1/T, the activation energy, E_ac_, can be determined for LSO and LSTO glasses. The straight lines in the figure represent the relaxation activation energies. The ac activation energy, E_ac_, of the LSO glass is the same as the one of the LSTO glass, where the value is 0.62 ± 0.01 eV for LSO and LSTO glasses. The activation energies are almost identical for both representations of the electrical modulus and the power law conductivity.

As shown in Figure 7, the maximum relaxation frequency of LSTO glass is higher than that of LSO glass at a given temperature, implying that the relaxation of the LSTO sample is more rapid than the relaxation of the LSO sample for an external field. The reason for the rapid relaxation of the LSTO glass may be due to the weaken bonds of structural network in the modified glass when the TeO_2_ is added in the lithium-silicate glass. 

## 4. Conclusions

The effect of adding tellurite to lithium-silicate glass is investigated by using DTA and impedance analyses from Cole−Cole, power law and modulus representations. 

The LSO and LSTO glasses synthesized by the rapid quenching method show that the tellurite modifies the structural network of LSO glass to be weakened. 

The observed dc and ac activation energies of the LSO and LSTO glasses are the same. The ac conductivity increases when TeO_2_ is added to the lithium-silicate glass. This increase of ionic conductivity has been attributed to an increase in number density for LSTO glass compared with the one for LSO glass. The modulus analysis confirms that the lithium ions of LSTO glass are trapped strongly in the potential wells.

## Figures and Tables

**Figure 1 materials-13-05232-f001:**
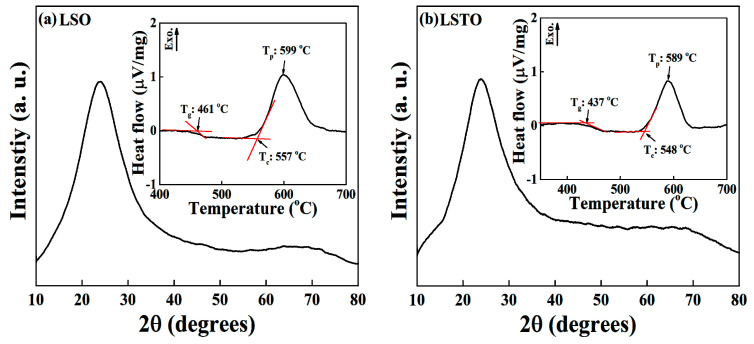
XRD patterns of the (**a**) Li_2_O-2SiO_2_ (LSO) and (**b**) 0.9 (Li_2_O-2SiO_2_)-0.1 TeO_2_ (LSTO) glasses at room temperature. The XRD patterns in Figure 1a,b were averaged for five times scan data with each scan 0.05-degree step for three seconds, in the scattering angle from 10 to 80 degrees. The broad XRD pattern is the typical glass characteristic scattered from the short ranging disordered network structure. Inset: DTA curves of (**a**) LSO and (**b**) LSTO glasses. The heating rate is 10 °C/min. We denote T_g_ for the glass transition temperature, T_c_ for the crystallization temperature and T_p_ for the exothermic peak temperature.

**Figure 2 materials-13-05232-f002:**
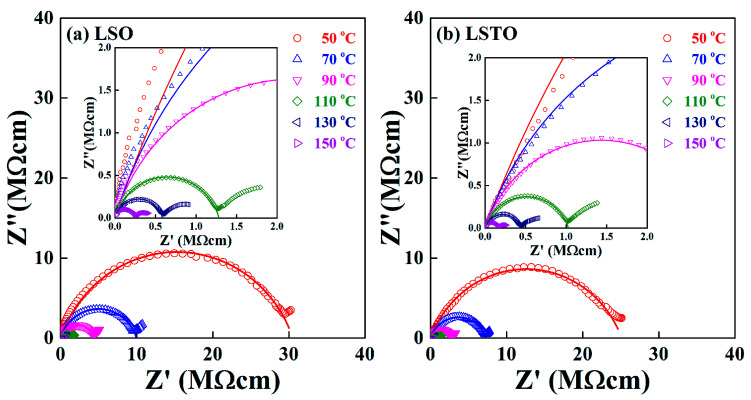
Complex impedance Cole−Cole plot of (**a**) LSO and (**b**) LSTO glasses, measured at various temperatures. Inset shows the extended complex impedance patterns at a high frequency regime.

**Figure 3 materials-13-05232-f003:**
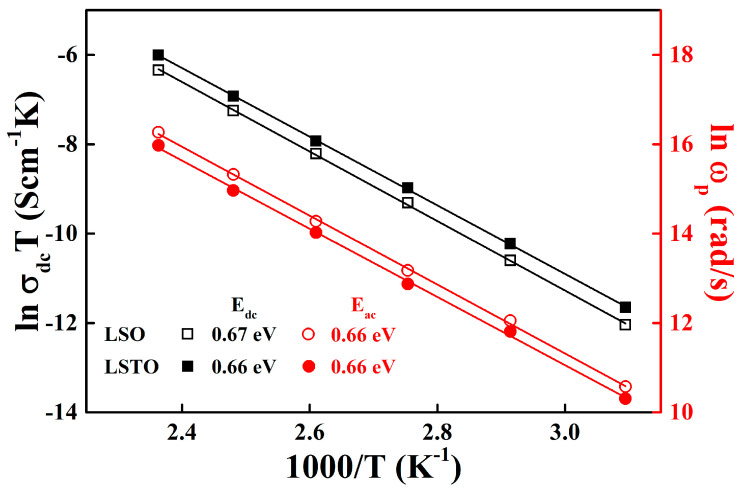
Temperature dependence of the dc conductivity (black: left vertical axis) and the conductivity relaxation peak frequency, ω_p_, (red: right vertical axis) obtained from the complex impedance Cole−Cole analysis for LSO (opened symbols) and LSTO (closed symbols) glasses. The solid lines are the least-squares straight-line fits to the data.

**Figure 4 materials-13-05232-f004:**
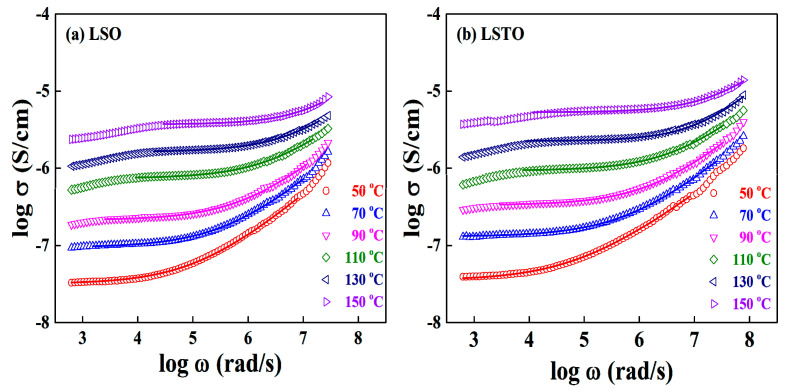
At various temperatures, frequency spectra of the real conductivity σ(ω) for (**a**) LSO and (**b**) LSTO-glasses. To obtain the solid lines, we do fit the experimental data by using a Jonscher’s power law in Equation (2).

**Figure 5 materials-13-05232-f005:**
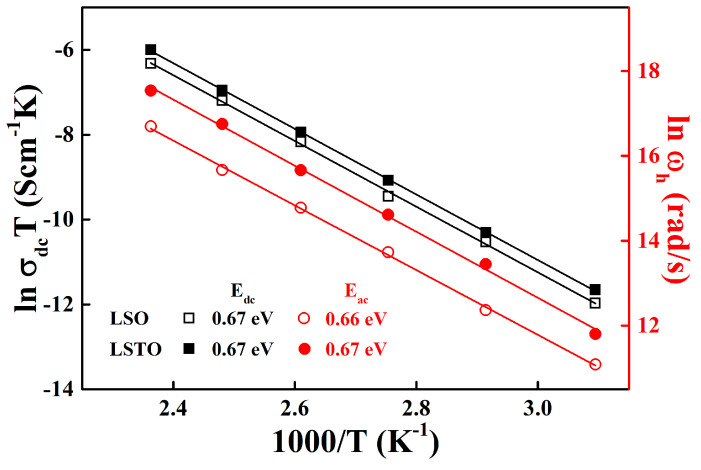
Temperature dependence of the dc conductivity (black: left vertical axis) and the hopping frequency $\omega_{h}$ (red: right vertical axis) obtained from the power-law analysis for LSO (opened symbols) and LSTO (closed symbols) glasses. Reciprocal temperature dependence of hopping frequency for two different compositions of LSO and LSTO glasses obtained from a fit to $\sigma \left(\omega_{h}\right)=2\sigma_{\mathrm{dc}}$. The red lines indicate that $\omega_{h}$ obeys the Arrhenius relation. The solid lines are the least-squares straight-line fits to the data.

**Figure 6 materials-13-05232-f006:**
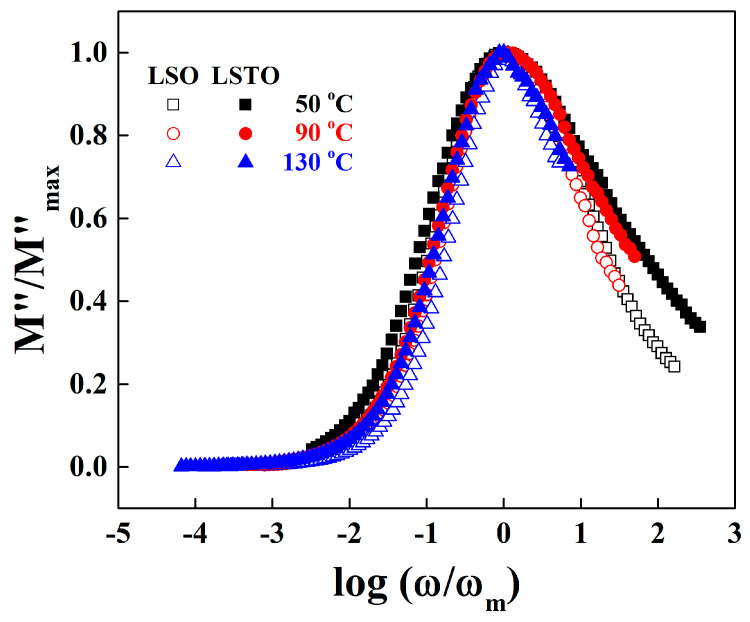
Normalized plot of dielectric modulus against normalized frequency for LSO (opened symbols) and LSTO (closed symbols) glasses at various temperatures.

**Figure 7 materials-13-05232-f007:**
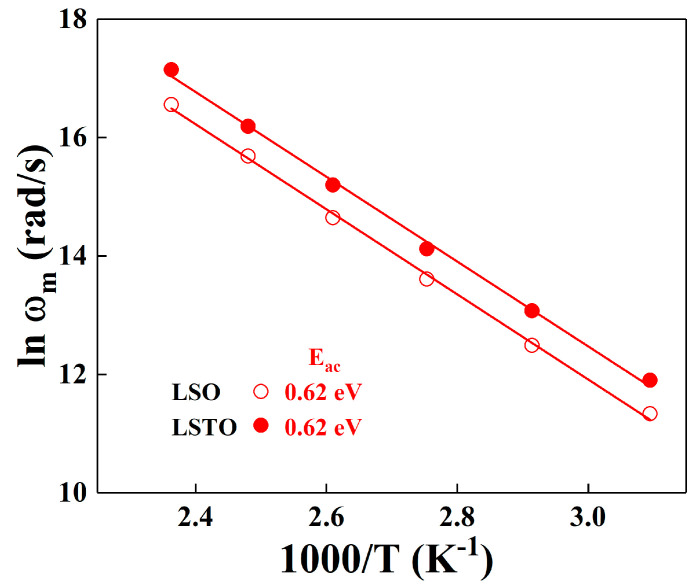
Temperature dependence of the ac conductivity, which is fit from the dielectric relaxation peak frequency ω_m_, obtained from the modulus analysis for LSO (opened symbols) and LSTO (closed symbols) glasses. The solid lines are the least-squares straight-line fits to the data.

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
