# Peer review of "Insight into Electrical and Dielectric Relaxation of Doped Tellurite Lithium-Silicate Glasses with Regard to Ionic Charge Carrier Number Density Estimation"

_materials, 2020, doi:10.3390/ma13225232_

Round 1

Reviewer 1 Report

This work is devoted to fabrication and investigation of electrical properties of doped tellurite lithium-silicate glasses. Conductive solid electrolytes for lithium-air and lithium-polymer batteries is very important topic in terms of environmental pollutions. The study is well designed and represented. Some comments are listed below:

1) p. 2, line 50. 10-5.75 S/cm better to write in a more typical way (power should be integer)

2) Comparison of Li-based glass with polymer electrolytes and Nasicon should be provided

3) More detailed comparison with other ceramic glasses in intoduction should be provided

4) the application of tellurium should be explained in introduction

5) fig. 1, XRD data is not processed

6) in fig. 1 and 2 better situate a) and b) in horizontal way 

Author Response

Reviewer1: Comments and Suggestions for Authors

This work is devoted to fabrication and investigation of electrical properties of doped tellurite lithium-silicate glasses. Conductive solid electrolytes for lithium-air and lithium-polymer batteries is very important topic in terms of environmental pollutions. The study is well designed and represented. Some comments are listed below:

1) p. 2, line 50. 10-5.75 S/cm better to write in a more typical way (power should be integer)

Reply;

As per your suggestions, we have changed the statements in the introduction section of the line numbers 49-50 in page 2 as following:

They reported that their system showed a lithium ionic conductivity of 10-5.75 S/cm at 150 ℃.

à They reported that their system showed a lithium ionic conductivity 1.78 of x 10-6 S/cm at 150 ℃.

2) Comparison of Li-based glass with polymer electrolytes and Nasicon should be provided

Reply;

We have rewritten and added the statements in the introduction section on pages 1 and 2 between the lines 26 and 69 as following:

The lithium-ion conductors provide tremendous opportunities in the development of solid-state lithium-ion and lithium-air batteries for vehicle applications. Lithium silicate glasses are considered promising nominees for applications as solid electrolyte in lithium ion batteries. Ionic solid electrolytes may be potential candidates for high voltage batteries due to their high electrochemical stability and safety. The remarkable research is still under progress to improve the compatibility between solid electrolytes and reactive electrodes [1–6].

Although, the lithium silicate glass is one of the candidate materials for the solid electrolyte in lithium ion batteries but it has a low ionic conductivity. The use of the glass-ceramic solid electrolytes leads to the development of a bulk-type all solid-state lithium ion batteries with excellent cycling performance. Glass and glass-ceramics have many advantages over their crystalline counterparts such as isotropic physical features, the absence of grain boundaries, and continuous composition changes [7]. It is a known feature of ion conducting glasses that the network dynamics has a significant impact on the ion dynamics [8–10].

One method for increasing the ionic conductivity is the mixing of different types of glass formers, which is known as the mixed glass former effect (MGFE). By mixing glass formers such as B2O3 with SiO2 at constant molar amount of modifying alkali oxide such as Li2O, the free-energy landscape for the ion migration changes. As a consequence, both the energy levels for the residence sites of the Li+ ions and the saddle point energies for ion jumps between neighboring non-bridging oxygens (NBOs) sites are affected [11–14]. In some cases, positive MGFE can produce advantageous changes in glass properties important to their application as solid electrolytes [15].

Recently, Rodrigues et al. synthesized the glass samples of the 0.30Li2O-0.70(xTeO2-(1-x)SiO2) and investigate the electrical conductivity behaviors in terms of the MGFE between TeO2 and SiO2 [16]. The electrical conductivity increased as the increase in the tellurium oxide content to x = 0.25 in the ternary glass system. They reported that their system showed a lithium ionic conductivity of 10-5.75 S/cm at 150 ℃. In addition, tellurium oxide-based glasses are of scientific and technological interest on account of their low melting temperature, high refractive index, high dielectric constant, high infrared transmission, and low phonon energy [17].

We observe the consistent results that the electrical conductivity of the 0.9(Li2O-2SiO2) – 0.1TeO2 (LSTO) glass increases slightly in comparison with the conductivity of the Li2O-2SiO2 (LSO) glass. Although a MGFE is a good description for increment of electrical conductivity when TeO2 is added to the Li2O-SiO2 glass, the further investigation is needed to answer for the fundamental questions on the origin of ionic conduction mechanism how many conducting ions are participating in a doped disordered system.

Recently, Rim et al. showed that the modified fractional Rayleigh equation is very useful to understand the fundamental mechanism for ionic transport in glasses [18]. Correspondingly, we can estimate the ionic carrier number density in the glass system, which is important to understand directly for the electrical conductivity, relating to the available sites in the glasses.

In a previous paper, we found that the partial Li+ ions in the lithium silicate glasses participated on the ac conduction through the non-bridging oxygens (NBOs) sites, which were created via breaking of SiO4 tetrahedral units with different numbers of NBO atoms per a silicon atom [14].

In this paper, we aim to find out what are the effects of doped tellurium oxide for Li ionic conductivity in the silicate glasses by comparing the electrical impedance of LSO and LSTO glasses. We analyze the characteristics of exponents of Cole-Cole plot, power-law and modulus representations to unveil the differences in the ionic conductivity between LSO and LSTO glasses.

à Understanding the lithium ion conduction provides a broad application opportunity ranging from the small size of portable devices to large size of vehicles and electric energy storage systems as a lithium ion secondary battery consisted of cathode, anode and electrolyte. Concerning the electrolyte of lithium ion battery, most of the cases, the host material of lithium ion conduction can be a liquid, polymer or solid. Up to now, in the sense of applications, liquid and polymer have been used widely due to the easy fabrication process and high ionic conductivity. Nonetheless, with the enormous increase on the demand of lithium ion battery, higher safety and longer lifetime of the battery has been required. All solid state batteries, where the components consisting of the batteries are solid, have turned out to be the best candidates to fulfill those conditions. And thus studies have been concentrated on developing solid state materials for the use of cathode, anode and electrolyte of lithium ion battery [1-6].

Regard to the solid state electrolytes, material types can be a crystal or a glass. Among these, the glass material has its own advantages characterized by the possible selection of various atomic species, easy control of synthesizing temperature by choosing proper glass former, easy variation of doping rates and types for improving electric properties, and lack of directional malfunctioning [7]. Lithium silicates, as a form of ceramics, have long been studied because of such useful applications as dental materials and coating binders. Meanwhile, lithium silicate glasses with a high concentration of lithium have been recognized to be used for solid state electrolyte of lithium ion battery. One of the barriers to overcome in using solid state materials for electrolyte is a relatively low electrical conductivity.

There have been continuous efforts to develop crystalline solid electrolytes based on the traditional materials. Those are lithium superionic conductor (LISICON), normally refers to the chemical formula Li2+2xZn1-xGeO4 and LISICON-like such as Li10MP2S12 (M = Si, Ge, Sn), exhibiting ionic conductivity of about 10-6 and 10-2 Scm-1 at room temperature, respectively [8]. In the sense of extending conducting ions, sodium superionic conductor (NASICON) with the normal chemical formula Na1+xZr2SixP3-xO12 and NASICON-like such as Li1.3Al0.3Ti1.7(PO4)3, are also widely studied materials, but their ionic conductivities are one order of lower compared with those of LISICON and LISICON-like materials due to the heavier mass of conducting ion [8].

Solid polymers of polyesters, polyamines and polysiloxane are also candidates for electrolytes. Ions are usually conduct through the polymer chains. Polymers are easy to process and advantageous for large scale production, high plasticity and elasticity. Considering the low limit of conductivity 10-4 Scm-1 for solid electrolytes, most polymer electrolytes are within this range of conductivity but the high limit of polymer is confined to about 10-3 Scm-1 at room temperature [9].  

Concerning amorphous solid electrolytes under development, there can be two types of non-oxide and oxide ceramic glasses. Li2S-GeS2, Li2S-P2S5 with conductivity about 10-4 Scm-1 and Li2S-SiS2-Li3PO4, Li2S-SiS2-Li4SiO4 with conductivity about 10-3 Scm-1 are typical ceramic glass electrolytes which have high ionic conductivities comparable to those of LISICON and NASICON series [10,11].

Tellurium is one of the chalcogens and, when it is doped or added to a host, the mechanical, electrical, optical properties of material can be controlled. It vulcanizes rubber, changes electronic current in semiconductor and tints crystal or glass color. In the form of its oxide, the tellurite acts as a glass former and leads tellurite mixed materials to be disordered network structure [12]. Therefore, as the case in this work, the addition of tellurium or tellurite in an oxide material allows us to adjust both electrical conductivity and glass forming ability.

Recently, Rodrigues et al. investigated the electrical conductivity of 0.30Li2O-0.70(xTeO2-(1-x)SiO2) glass [13] and found that the electrical conductivity increased as the increase in the tellurium oxide content to x = 0.25. They reported that their system showed a lithium ionic conductivity 1.78 of x 10-6 S/cm at 150 ℃.

We observe the consistent results that the electrical conductivity of the 0.9(Li2O-2SiO2) – 0.1TeO2 (LSTO) glass increases in comparison with the conductivity of the Li2O-2SiO2 (LSO) glass. And the further investigation is needed to answer for the fundamental questions on the origin of ionic conduction mechanism.

Recently, we showed that, with the modified fractional Rayleigh equation, the ionic carrier number density in the glass system can be estimated [14].

In this paper, we aim to find out the effects of doped tellurium oxide for Li ionic conductivity in the silicate glasses by comparing the electrical impedance of LSO and LSTO glasses. We analyze the electrical characteristics using Cole-Cole plot, power-law and modulus representations to unveil the differences in the ionic conductivity between LSO and LSTO glasses.

As per your suggestions, we have added the statements in the introduction section as following:

à There have been continuous efforts to develop crystalline solid electrolytes based on the traditional materials. Those are lithium superionic conductor (LISICON), normally refers to the chemical formula Li2+2xZn1-xGeO4 and LISICON-like such as Li10MP2S12 (M = Si, Ge, Sn), exhibiting ionic conductivity of about 10-6 and 10-2 Scm-1 at room temperature, respectively. In the sense of extending conducting ions, sodium superionic conductor (NASICON) with the normal chemical formula Na1+xZr2SixP3-xO12 and NASICON-like such as Li1.3Al0.3Ti1.7(PO4)3, are also widely studied materials, but their ionic conductivities are one order of lower compared with those of LISICON and LISICON-like materials due to the heavier mass of conducting ion.

Solid polymers of polyesters, polyamines and polysiloxane are also candidates for electrolytes. Ions are usually conduct through the polymer chains. Polymers are easy to process and advantageous for large scale production, high plasticity and elasticity. Considering the low limit of conductivity 10-4 Scm-1 for solid electrolytes, most polymer electrolytes are within this range of conductivity but the high limit of polymer is confined to about 10-3 Scm-1 at room temperature.  

3) More detailed comparison with other ceramic glasses in introduction should be provided

Reply;

As per your suggestions, we have changed and added the statements in the introduction section as following:

à Concerning amorphous solid electrolytes under development, there can be two types of non-oxide and oxide ceramic glasses. Li2S-GeS2, Li2S-P2S5 with conductivity about 10-4 Scm-1 and Li2S-SiS2-Li3PO4, Li2S-SiS2-Li4SiO4 with conductivity about 10-3 Scm-1 are typical ceramic glass electrolytes which have high ionic conductivities comparable to those of LISICON and NASICON series.

4) the application of tellurium should be explained in introduction

Reply;

As per your suggestions, we have changed and added the statements in the introduction section as following:

à Tellurium is one of the chalcogens and, when it is doped or added to a host, the mechanical, electrical, optical properties of material can be controlled. It vulcanizes rubber, changes electronic current in semiconductor and tints crystal or glass color. In the form of its oxide, the tellurite acts as a glass former and leads tellurite mixed materials to be disordered network structure [Stanworth, J. E. Tellurite Glasses, Nature 1952, 169, 581-582.]. Therefore, as the case in this work, the addition of tellurium or tellurite in an oxide material allows us to adjust both electrical conductivity and glass forming ability.

5) fig. 1, XRD data is not processed

Reply;

We add some sentences in the results and discussion between the line 102 and 103 and in figure cation in Fig.1.

à  The XRD patterns in Fig. 1(a) and 1(b) were averaged for five times scan data with each scan 0.05 degree step for three seconds, in the scattering angle from 10 to 80 degree.

Added sentences; The figure exhibits the broad peaks of amorphous phase. The broad XRD pattern is the typical glass characteristics scattered from the short ranging disordered network structure.

6) in fig. 1 and 2 better situate a) and b) in horizontal way 

Reply;

We correct and replace the Fig. 1(a) and 1(b) between the line 110 and 111 as following:

à

We correct and replace the Fig. 2(a) and 2(b) between the line 126 and 127 as following:

à

We have added the more references in the introduction section on page 3 and have arranged the references in orders as following:

à8. Bachman, J. C.; Muy, S.; Grimaud, A.; Chang, H.-H.; Pour, N.; Lux, S. F.; Paschos, O.; Maglia, F.; Lupart, S.; Lamp, P.; Giordano, L.; and Yang, S.-H. Inorganic Solid-State Electrolytes for Lithium Batteries: Mechanisms and Properties Governing Ion Conduction, Chem. Rev. 2016, 116, 140−162. doi:10.1021/acs.chemrev.5b00563.

  1. Zhou, D.; Shanmukaraj,D.; Tkacheva, A.; Armand, M.; Wang, G. Polymer Electrolytes for Lithium-Based Batteries: Advances and Prospects, Chem 2019, 5, 2326-2352, September, 12, doi: 10.1016/j.chempr 2019.05.009.
  2. Trevey, J. E.; Jung, Y. S.; Lee, S.-H. High Lithium Ion Conducting Li2S–GeS2–P2S5 Glass–Ceramic Solid Electrolyte with Sulfur Additive for all Solid-State Lithium Secondary Batteries, Electrochem. Acta 2011, 56, 42423-4247. doi:10.1016/j.electacta.2011.01.086.

11.Takada, K.; Aotani, N.; Kondo, S. Electrochemical Behaviors of Li+ Ion Conductor, Li3P04-Li2S-SiS2, J. Power Sources 1993, 43-44, 135-141.

  1. Stanworth, J. E. Tellurite Glasses, Nature 1952, 169, 581-582.

Reviewer 2 Report

In this manuscript, the authors have presented an experimental study to investigate the role of tellurite on lithium-silicate (LSTO) glass. It has been shown that the increase in lithium-ion leads to enhanced electrical conductivity in the LSTO glass. Further, the ionic hopping and relaxation process has been explained using the Cole-Cole power law. The work presented in this manuscript is scientifically sound and well written. However, the introduction section is short and does not offer a clear motivation and a detailed discussion of the existing literature.

Similarly, the conclusion section only contains the summary of results, no discussion, and a minimal future outlook. Furthermore, the results should be discussed in more detail, and connections to the previous works should also be addressed. One minor observation is there a specific need to show two colored axes in Fig. 7.

Overall, the manuscript is not appropriate for a review article in its present form. If the authors revise the manuscript and consider the issues mentioned above, I may recommend it for publication in Materials.

Author Response

We submit as a file

Reviewer 3 Report

 Minor point: Please revise that all unusual abbreviations are appropriatedly defined when it appear in text for first time (i.e. dc or ac in Abstract.

MANDATORY: Re-write Conclusions as a short and clear paragraph enhancing the essential contribution of the work.

They are not an additional summary of resulst and discusions!!!!

Author Response

We submit as a file

Round 2

Reviewer 3 Report

All is now OK